# How to tell when a clustering is (approximately) correct using convex relaxations

**Marina Meila**[*]
University of Washington
Seattle, WA 98195
mmp@stat.washington.edu

## Abstract

We introduce the Sublevel Set (SS) method, a generic method to obtain sufficient guarantees of near-optimality and uniqueness (up to small perturbations) for a clustering. This method can be instantiated for a variety of clustering loss functions for which convex relaxations exist. Obtaining the guarantees in practice amounts to solving a convex optimization. We demonstrate the applicability of this method by obtaining distribution free guarantees for K-means clustering on realistic data sets.

## 1 Introduction

This paper proposes a framework for providing theoretical guarantees for clustering, without making (untestable) assumptions about the data generating process. The main question we address is: can a user tell, with no prior knowledge, if the clustering $\mathcal{C}$ returned by a clustering algorithm is meaningful? This is the fundamental problem of cluster validation. We build on the idea of [Mei06] who proposed that: to be meaninful, a clustering $\mathcal{C}$ must be both "good" and *the only* "good" clustering of the data $\mathcal{D}$, up to small perturbations. Such a clustering is called *stable*. Data that contains a stable clustering is said to be *clusterable*.

In this paper, we adopt the *loss-based* clustering framework, where for a given number of clusters $K$, the best clustering of the data is the one that minimizes a *loss function* $\mathrm{Loss}(\mathcal{D}, \mathcal{C})$. This framework includes K-means, K-medians, graph partitioning, as well as model-based clustering (by letting the loss function be the data log-likelihood.[2]). Consequently, a good clustering $\mathcal{C}$ has low $\mathrm{Loss}(\mathcal{D}, \mathcal{C})$, in a way that will become precise later. Supposing that it is possible to find a good clustering, the challenge is to verify that $\mathcal{C}$ is stable without enumerating all the other clusterings. Hence this work will show how to obtain theorems like the following.

**Stability Theorem** (∗). *Given a clustering $\mathcal{C}$ of data set $\mathcal{D}$, a function $\mathrm{Loss}(\mathcal{D}, \mathcal{C})$, and technical conditions $\mathcal{T}$, there is an $\epsilon$ such that $d(\mathcal{C}, \mathcal{C}') \leq \epsilon$ whenever $\mathrm{Loss}(\mathcal{D}, \mathcal{C}') \leq \mathrm{Loss}(\mathcal{D}, \mathcal{C})$.*

If we can show this for a clustering $\mathcal{C}$, it means that $\mathcal{C}$ captures structure existing in the data, thus it is meaningful. It should also be evident, that it is not possible to obtain such guarantees in general; they can only exist for clusterable data, as illustrated in Figure 1.

The rest of the paper will describe a generic method for obtaining technical conditions $\mathcal{T}$ and stability theorems such as **Stability Theorem** (∗) above for loss-based clustering (section 2). We will illustrate the working of this method for the K-means cost function (sections 3 and 6), and will present further instantiations in Section 4 and related work in Section 5. Section 7 concludes the paper. While the idea of using stability as in (∗) was introduced by [Mei06] who used spectral bounds, the contributions of this paper are (1) to greatly expand the scope of [Mei06] from spectral bounds to general tractable

---

[*]www.stat.washington.edu/mmp

[2]Restrict to hard clusterings only.

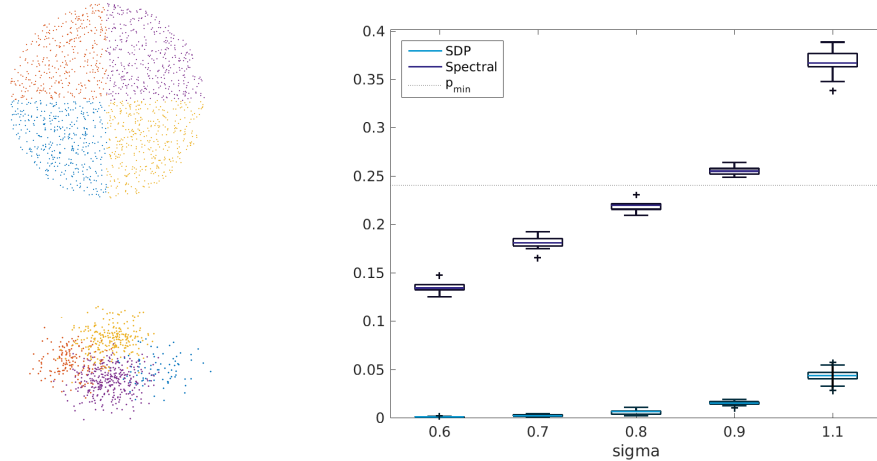

Figure 1: Top,left: A data set that is not clusterable: the clustering shown is nearly optimal, but not stable. Bottom,left: A clusterable data set, described in the Experiments section 6, ($n = 1024$) with a stable and nearly optimal clustering $\mathcal{C}$ w.r.t the K-means loss. The spherical clusters have $\sigma = 0.9$, the cluster centers are $4\sqrt{2} \approx 5.67$ apart; the method described in Section 3 guarantees than all clusterings of this data set that are at least as "good" as $\mathcal{C}$ differ from it in at most $1.44\%$ of the points. Right: $\varepsilon$ and $\varepsilon_{Sp}$ for $K = 4$, $n = 1024$ and various $\sigma$ values (over 10 replications). The values of $\varepsilon_{Sp}$ exceeding $p_{\min}$ are not valid BTRs.

relaxations and to a much wider class of clustering problems, (2) to obtain new results in the case of the K-means loss by using the Semidefinite Programming (SDP) relaxations of K-means, and (3) to demonstrate that these are much tighter than the previous ones.

## 2 Proving stability via convex relaxations

### 2.1 Preliminaries and definitions

**Representing clusterings as matrices**    Let $\mathcal{D} = \{x_1, \ldots x_n\}$ be the data to be clustered. We make no assumptions about the distribution of these data. A *clustering* $\mathcal{C} = \{C_1, \ldots C_K\}$ of the dataset $\mathcal{D}$ is a partition of the indices $\{1, 2, \ldots n\} := [n]$ into $K$ non-empty mutually disjoint subsets $C_1, \ldots C_K$, called *clusters*. Let $n_k = |C_k|$ for $k \in [K]$; $\sum_{k=1}^{K} n_k = n$, $p_{\min} = \min_{k \in [K]} n_k/n$, $p_{\max} = \max_{k \in [K]} n_k/n$. We denote by $\mathbf{C}_K$ the space of all clusterings with $K$ clusters. A clustering can be represented by an $n \times n$ *clustering matrix* $X$ defined as

$$X_{ij} = \begin{cases} 1/n_k & \text{if } i, j \in C_k \text{ for some } k \in [K] \\ 0 & \text{otherwise} \end{cases} \quad , \quad X = [X_{ij}]_{i,j=1}^{n}. \tag{1}$$

The following proposition lists the properties of the matrix $X$. Its proof as well as all other proofs can be found in the supplement.

**Proposition 1.** *For any clustering $\mathcal{C}$ of $n$ data points, the matrix $X$ defined by* (1) *has elements $X_{ij}$ in $[0, 1]$; $\operatorname{trace} X = K$, $X\mathbf{1} = \mathbf{1}$, where $\mathbf{1} = [1 \ldots 1]^T$, $\|X\|_F^2 = \operatorname{trace} X^T X = K$. Moreover, $X \succeq 0$, i.e. $X$ is a positive semidefinite (PSD) matrix (i.e. $X$ symmetric and $x^T X x \geq 0$ for all vectors $x$).*

To distinguish a clustering matrix $X$ from other $n \times n$ symmetric matrices satisfying Proposition 1, we sometimes denote the former by $X(\mathcal{C})$.

**Measuring the distance between two clusterings**    The *earthmover's distance* (also called the *misclassification error distance*) between two clusterings $\mathcal{C}, \mathcal{C}'$ over the same set of $n$ points is

$$d^{EM}(\mathcal{C}, \mathcal{C}') = 1 - \frac{1}{n} \max_{\pi \in \mathbb{S}_K} \sum_{k=1}^{K} |C_k \cap C'_{\pi(k)}|, \tag{2}$$

where $\pi$ ranges over the set of all permutations of $K$ elements $\mathbb{S}_K$, and $\pi(k)$ indexes a cluster in $\mathcal{C}'$. This definition can be generalized to clusterings with different numbers of clusters and to unequally weighted data points.

## 2.2 Sublevel set problems and outline of the method

**Losses and convex relaxations**   A *loss function* $\mathrm{Loss}(\mathcal{D}, \mathcal{C})$ (such as the K-means or K-medians loss), specifies what kind of clusters the user is interested in, via the optimization problem below.

$$\text{Clustering problem:} \quad L^{\mathrm{opt}} \;=\; \min_{\mathcal{C} \in \mathbf{C}_K} \mathrm{Loss}(\mathcal{D}, \mathcal{C}), \quad \text{with solution } \mathcal{C}^{\mathrm{opt}} \qquad (3)$$

As most loss functions require a number of clusters $K$ as input, we assume that $K$ is fixed and given. In Section 7 we return to the issue of choosing $K$. The majority of interesting loss functions result in combinatorial optimization problems (3) known to be hard in the worst case.

Let $\mathcal{X}$ be a convex set in the Euclidean space, such that $\mathcal{X} \supset \{X(\mathcal{C}), \mathcal{C} \in \mathbf{C}_K\}$. If we extend $\mathrm{Loss}(\mathcal{D}, \mathcal{C})$ to $\mathrm{Loss}(\mathcal{D}, X)$ convex in $X$ for all $X \in \mathcal{X}$, then the problem

$$L^* \;=\; \min_{X \in \mathcal{X}} \mathrm{Loss}(\mathcal{D}, X), \quad \text{with solution } X^* \qquad (4)$$

is a *convex relaxation* of the problem (3). In the above, the representation $X(\mathcal{C})$ can be the one defined in (1), or a different injective mapping of $\mathbf{C}_K$ to an Euclidean space. Because $\mathcal{X} \supset \mathbf{C}_K$, $L^* \leq L^{\mathrm{opt}}$ and $X^*$ is generally not a clustering matrix. On the other hand, finding $X^*$ is typically tractable, while finding $\mathcal{C}^{\mathrm{opt}}$ is not.

Convex relaxations for clustering have received considerable interest. For graph partitioning problems, [XJ03] introduced two SDP relaxations, [HS11] proposed non-convex but tight relaxations to a class of graph partitioning problems, while [RMH14] proposed a *continuous relaxed balanced cut* convex problem that depends on a submodular set function $S$. By judiciously choosing $S$, one obtains tight relaxations to various classes of graph cut problems. Correlation clustering, a graph clustering problem appearing in image analysis, has been given an SDP relaxation in [Swa04, AS16b]. For community detection under the Stochastic Block Model [HLL83] several SDP relaxations have been recently introduced [CX14, VOH14, JHDF16] as well as Sum-of-Squares relaxations for finding hidden cliques [DM15]. For centroid based clustering, we have LP relaxations for K-medians [CG99] and K-means [ACKS15] and more recent, tighter relaxations via SDPs in [ABC$^+$14]. The SDP relaxations [ABC$^+$14, IMPV15b] for K-means have guarantees under the specific generative model called the Stochastic Ball Model [IMPV15a]. For hierarchical clustering in the cost-based paradigm introduced by [Das16], we have LP relaxations introduced by [RP16, CG99, CC16]. We also mention the related area of *convex clustering* [BH07] where loss functions are designed to be convex. These can often be seen as relaxations to standard K-center or exemplar based losses.

**The sublevel set method**   Now we show how to use an existing relaxation to obtain guarantees of the form $(*)$ for clustering. Given a Loss, its clustering problem (3), and a convex relaxation (4) for it we proceed as follows:

Step 1  Use the convex relaxation to find a set of good clusterings that contains a given $\mathcal{C}$ . This set is $\mathcal{X}_{\leq l} = \{X \in \mathcal{X}, \mathrm{Loss}(\mathcal{D}, X) \leq l\}$, the *sublevel set* of Loss, at the value $l = \mathrm{Loss}(\mathcal{D}, \mathcal{C})$. This set is convex when Loss is convex in $X$.

Step 2  Show that if $\mathcal{X}_{\leq l}$ has sufficiently small diameter, then all clusterings in it are contained in the ball $\{\mathcal{C}', d^{\overline{EM}}(\mathcal{C}, \mathcal{C}') \leq \epsilon\}$. We will call this $\epsilon$ a *Bound from Tractable Relaxation (BTR)*.

In more detail, consider a dataset $\mathcal{D}$, with a clustering $\mathcal{C} \in \mathbf{C}_K$. If for the given Loss a convex relaxation exists, let its feasible set be $\mathcal{X}$, and let $X(\mathcal{C})$ be the image of $\mathcal{C}$ in $\mathcal{X}$. To accomplish Step 1, we modify (4) into the optimization problem below, which we call *Sublevel Set (SS)* problem.

$$\text{SS} \quad \epsilon'_1 \;=\; \max_{X' \in \mathcal{X}} \|X(\mathcal{C}) - X'\|, \quad \text{s.t. } \mathrm{Loss}(\mathcal{D}, X') \leq \mathrm{Loss}(\mathcal{D}, \mathcal{C}). \qquad (5)$$

In the above, the norm $\|\,\|$ can be chosen conveniently; in this paper it will be the Frobenius norm. The feasible set for (5) is $\mathcal{X}_{\leq \mathrm{Loss}(\mathcal{D}, \mathcal{C})}$, a convex set. The tractability of (5) depends on the objective $\|X(\mathcal{C}) - X'\|$, which generally is *not concave*. In the next sections it will be shown that the mapping $X(\mathcal{C})$ in (1) always leads to tractable SS problems, and we will present other examples of such

mappings. When SS is tractable, by solving it we obtain that $||X(\mathcal{C}') - X(\mathcal{C})|| \leq \epsilon'$ for all clusterings $\mathcal{C}'$ with $\text{Loss}(\mathcal{D}, \mathcal{C}') \leq \text{Loss}(\mathcal{D}, X(\mathcal{C}))$. Thus, SS finds a ball centered at $\mathcal{C}$ that contains all the good clusterings. Here "good" means "at least as good as $\mathcal{C}$" w.r.t. Loss, but any sublevel set can be considered, even for levels lower than $\text{Loss}(\mathcal{D}, \mathcal{C})$. The radii of these sublevel sets tell us how clusterable the data is, in a norm that depends on the relaxation $\mathcal{X}$.

$||X(\mathcal{C}') - X(\mathcal{C})||$ could be considered a distance between partitions, but this "distance" is less intuitive, and has the added disadvantage that it depends on the mapping $X$ used. In Step 2, we transform the bound $\varepsilon'_1$ into a bound for the earthmover's distance $d^{EM}$. In the next sections we provide examples and sufficient conditions when this is possible by existing methods.

## 3 BTR bounds for the K-means loss

**The K-means clustering paradigm**    In K-means clustering, the data points are vectors $x_{1:n} \in \mathbb{R}^d$. The objective is to minimize the *squared error loss*, also known as the *K-means loss*

$$\text{Loss}(\mathcal{D}, \mathcal{C}) = \sum_{k=1}^{K} \sum_{i \in C_k} ||x_i - \mu_k||^2, \quad \text{with } \mu_k = \frac{1}{n_k} \sum_{i \in C_k} x_i, \quad \text{for } k \in [K]. \tag{6}$$

If one substitutes the expressions of the centers $\mu_{1:K}$ into Loss, one obtains a function of the matrix $X$ and the *squared distances matrix $D$*.

$$\text{Loss}(\mathcal{D}, \mathcal{C}) \equiv \text{Loss}(D, X(\mathcal{C})) = \frac{1}{2} \langle D, X \rangle, \tag{7}$$

where

$$D = [D_{ij}]_{i,j \in [n]}, \quad D_{ij} = ||x_i - x_j||^2 \tag{8}$$

and $\langle A, B \rangle$ denotes the Frobenius scalar product $\langle A, B \rangle \overset{def}{=} \text{trace}(A^T B)$. The norm $||x||$ denotes the Euclidean norm of $x$. Finding the best clustering of a data set $\mathcal{D}$ is thus equivalent to solving the following optimization problem [ABC+14], which we will refer to as the *K-means problem*.

$$\min_{\mathcal{C} \in \mathbf{C}_K} \langle D, X(\mathcal{C}) \rangle \tag{9}$$

**A SDP relaxation for the K-means problem**    The K-means loss is hard to optimize in general, because of the presence of local minima. But, since K-means is one of the most widely used and well studied methods for finding groups in data, several tractable relaxations for the K-means problem have been developed. [DH04, DGK04] introduced a spectral relaxation, [ABC+14] introduced two convex relaxations, one resulting in a Linear Program (LP), the other in a Semi-Definite Program (SDP). We present the latter here.

$$\min_{X \in \mathcal{X}} \langle D, X \rangle \text{ s.t. } \mathcal{X} = \{X \succeq 0, \text{ trace } X = K, X\mathbf{1} = \mathbf{1}, X_{ij} \geq 0 \text{ for } i, j \in [n]\} \subset \mathbb{R}^{n \times n}. \tag{10}$$

By Proposition 1, $X(\mathcal{C}) \in \mathcal{X}$ for all $\mathcal{C} \in \mathbf{C}_K$. In [ABC+14] it was shown that problem (10) is convex, and that it can be cast as a SDP. In general, $X^*$ the optimal solution of (10) may not be a clustering matrix. [ABC+14] showed that when data are sampled from well-separated discs, $X^*$ is a clustering matrix corresponding to the optimal clustering $\mathcal{C}^*$ of the data (and $\mathcal{C}^*$ assigns the points in each disk to a different cluster).

**A BTR for K-means**    In this section we explain how we obtain BTR bounds in the K-means clustering paradigm, by exploiting the relaxation (10).

We shall assume that a data set $\mathcal{D}$ is given, and that the user has already found a clustering $\mathcal{C}$ of this data set (by e.g. running the K-means algorithm). The user would like to know if: (a) is $\mathcal{C}$ optimal (in other words, is it the globally optimal solution to the non-convex problem (9))? and (b) could there be other clusterings of the data $\mathcal{D}$ that are very different from $\mathcal{C}$ but are similar or better w.r.t to Loss? As Figure 1 shows, the two questions are both important, if the goal of the clustering is to capture the structure of the data (i.e. the clustering) instead of minimizing the clustering Loss. We introduce the following SDP instantiating the generic SS problem (5).

$$(\text{SS}_{\text{Km}}) \quad \delta = \min_{X' \in \mathcal{X}} \langle X(\mathcal{C}), X' \rangle \quad \text{s.t.} \langle D, X' \rangle \leq \langle D, X(\mathcal{C}) \rangle \tag{11}$$

In the above, $X(\mathcal{C})$ is the clustering matrix of the known clustering, $\mathcal{X}$ is defined in (10) and $D$ is the squared distance matrix of $\mathcal{D}$. If $\mathcal{C}$ is given, the number of clusters is implicitly given by $K = |\mathcal{C}|$. Hence a user has all the information available for solving this SDP in practice. Note that the value $\delta$ is not given, but an output of the optimization algorithm solving (SS$_{\text{Km}}$).

Let us now examine what the optimal solution $X'$ and optimal value $\delta$ mean. First note that, because $||X||_F^2 = ||X'||_F^2 = K$ (by Proposition 1), $||X - X'||_F^2 = 2K - 2\langle X, X' \rangle$. Hence, the minimizer of $\langle X, X' \rangle$ is the same $X'$ that maximizes $||X - X'||_F$.

In comparison with (10), (SS$_{\text{Km}}$) adds an inequality constraint, thus restricting the feasible set of (10) to matrices $X'$ that have Loss no larger than the loss of $\mathcal{C}$. Both $X^*$ and $X(\mathcal{C})$ are feasible for (SS$_{\text{Km}}$), but clusterings with higher Loss than $\mathcal{C}$ are not. Hence, (SS$_{\text{Km}}$) finds among the feasible matrices $X'$ which have low loss, the one that is furthest away from $X(\mathcal{C})$. Typically, $X'$ is not in $\mathbf{C}_K$, and we are not interested in $X'$, but in how far it is from $X$. As Theorem 2 below will show, the optimum value $\delta$ in (SS$_{\text{Km}}$) determines this distance, measured in Frobenius norm, and $\delta \leq K$. Consequently, if the value $K - \delta$ is small enough, it implies that no good clusterings of the data can differ more than $K - \delta$ from $X(\mathcal{C})$. Therefore, our main result below states that when the value $\delta$ is near its maximum $K$, it controls the deviation from $\mathcal{C}$ of any other good clustering.

**Theorem 2.** *Let $\mathcal{D}$ be represented by its squared distance matrix $D$, let $\mathcal{C}$ be a clustering of $\mathcal{D}$, with $K, p_{\min}, p_{\max}$ defined as in Section 2.1, and let $\delta(\mathcal{C})$ be the optimal value of problem (SS$_{\text{Km}}$). Then, if $\varepsilon = (K - \delta(\mathcal{C}))p_{\max} \leq p_{\min}$, any clustering $\mathcal{C}'$ with $\mathrm{Loss}(\mathcal{C}') \leq \mathrm{Loss}(\mathcal{C})$ is at distance $d^{EM}(\mathcal{C}, \mathcal{C}') \leq \varepsilon$.*

We summarize the validation procedure below.

---

**Input** Data set with $D \in \mathbb{R}^{n \times n}$ defined as in (8), clustering $\mathcal{C}$ with $K$ clusters

**Preprocess** Calculate $p_{\min}, p_{\max}$, and clustering matrix $X(\mathcal{C})$.

1. Solve problem (SS$_{\text{Km}}$) numerically (by e.g. calling a SDP solver); let $\delta$ be the optimal value obtained.

2. Set $\epsilon' = K - \delta$ and $\epsilon = \epsilon' p_{\max}$.

3. **If** $\epsilon \leq p_{\min}$ **then**

   Theorem 2 holds: $\varepsilon$ is a BTR for $\mathcal{C}$.

   **else** no guarantees for $\mathcal{C}$ by this method.

---

Theorem 2 instantiates Stability Theorem ($*$). When the theorem's conditions hold, it provides a *certificate of stability* for $\mathcal{C}$. It is also evident that this can only happen if $\mathcal{C}$ is stable, which moreover is predicated on the data being clusterable.

## 4 For what other clustering paradigms can we obtain BTRs?

Now we show that the framework of Section 2 can be readily applied to several other clustering paradigms with very little extra work.

Define the following injective mappings of $\mathbf{C}_K$ into sets of matrices. The $X$ mapping is given in (1). The mapping $\tilde{X} \in \mathbb{R}^{n \times n}$ is given by $\tilde{X}_{ij} = 1$ if $i, j \in C_k$ for some $K$ and 0 otherwise. The mapping $Z \in \mathbb{R}^{n \times K}$ is given by $Z_{ij} = 1/\sqrt{n_k}$ if $i \in C_k$ for $k \in [K]$ and 0 otherwise.

**Theorem 3.** *Let* Loss *define a clustering paradigm that has a convex relaxation in which clustering $\mathcal{C}$ is mapped to one of the matrices $X, \tilde{X}, Z$ above. Then the following statements hold. (1) There exists a convex SS problem of the form $\delta = \min_{X' \in \mathcal{X}_{\leq l}} \langle X(\mathcal{C}), X' \rangle$ (and similarly for $\tilde{X}, Z$). (2) From the optimal value $\delta$ a BTR $\varepsilon$ can be obtained.*

*For the $X$ mapping, $\varepsilon = (K - \delta)p_{\max}$; for the $\tilde{X}$ mapping, $\varepsilon = \frac{\sum_{k \in [K]} n_k^2 + (n - K + 1)^2 + (K - 1) - 2\delta}{2p_{\min}}$; for the $Z$ mapping, $\varepsilon = (K - \delta^2/2)p_{\max}$. In all three cases, $\varepsilon$ is a BTR whenever $\varepsilon \leq p_{\min}$.*

This theorem can also be extended to cover weighted representations such as those used for graph partitioning in [MSX05]. Theorem 3 shows that getting bounds for a clustering paradigm does not depend directly on the Loss or clustering paradigm, but on the space of the convex relaxation. Moreover, somebody who already uses one of the above clustering relaxations cited would have very little to do to also obtain bounds.

Of the previously mentioned relaxations the $X$ mapping is used by [PW07, IMPV15b] for K-means in a SDP relaxations, by [RP16, CC16] for cost-based hierarchical clustering in an LP relaxation, and by [Swa04] for correlation clustering. The $\tilde{X}$ mapping is used by [CX14, JHDF16] for the Stochastic Block Model [HLL83], respectively by [VOH14] for the Degree-Corrected Stochastic Block Model [KN11]. The $Z$ mapping is used by the spectral relaxations [DH04, Mei06] for K-means. Finally, note that the relaxations in [HS11, RMH14] are not covered by Theorem 3, and obtaining BTR bounds from them looks both challenging and promising.

## 5   Related work

**Existing distribution free guarantees for clustering**   All the previous explicit BTR bounds we are aware of are based on spectral relaxations: [Mei06] gives a spectral bound for K-means and [MSX05, WM15] give BTR for graph partitioning. The work of [LGT14] relates the existence of good $r$-way graph partitioning to a large $K$-eigengap of the graph *normalized Laplacian*, where $r \geq K - 3K\delta$. More precisely, if $\lambda_{K+K\delta}/\lambda_K > c(\log K)^2/\delta^9$ then this partition is "better" than $\lambda_K/\delta^3 \times c'$ (for $c, c'$ unspecified). While these results are remarkable for their generality, they requires extremely large $\lambda_{K+K\delta}/\lambda_K$ to produce non-trivial bounds, no matter what $c, c'$ are; moreover, because $\lambda_{K+K\delta} \leq 1$, they also require $\lambda_K \ll 1$. In [PSZ15], a BTR bound for spectral clustering is given, which depends on unspecified constants.

**Algorithmic results under clusterability assumptions**   For finite mixtures, a series of remarkable results from the 2000's [Das00][AM05, VW04][DS07] and a few more recent ones [BMvL12][BWY14] established theoretical guarantees for recovery, together with tractable clustering algorithms. These papers are important because for the first time, recovery is not tied to maximizing the likelihood, but to the separation of the cluster centers, and to the relative sizes and spreads of the clusters. Recovery guarantees have been obtained also in *block-models* for network data, such as the *Stochastic Block Model (SBM)* [AS15, AS16a], *Degree-Corrected SBM (DC-SBM)*[QR13] and *Preference Frame Model (PFM)* [WM15].

Other researchers have shown that under resilience [BL09] or other clusterability assumptions on the data, one can find a (nearly) optimal clustering efficiently: [ACKS15, ABS12, ABV14, ABS12, ABS10, BL09, CFR15, BL16, BHW16, BLG14] and [Ben15], which offers a critical survey of this area, and underscores that the clusterability conditions are in general very restrictive. Similar results for graph clustering are given in e.g. [KVV00]. We have already mentioned the recent [ABC$^+$14, IMPV15b]. Our results are *complementary* to the work in this area. This work provides very strong evidence that if $\mathcal{D}$ is clusterable, a good clustering $\mathcal{C}$ is easy to find. This corroborates a large body of empirical evidence, including our own experiments in Section 6. In future work we aim to close the loop by providing end-to-end algorithms that both cluster data efficiently and give BTRs for the resulting $\mathcal{C}$. Second, our work *grounds* this area of research. By the SS method one could hope to prove the assumptions that (some of) the aforementioned algorithms are relying on, making them more relevant.

**Other work in unsupervised learning**   Recently, in [HM16] a PAC-like framework for unsupervised learning is proposed. Similar to our paper, the framework of [HM16] argues for the need of a *hypothesis class*, of an assumption that the data *fits the model class* (i.e., the $(k, \epsilon)$ decodability condition), and the use of problem specific tractable relaxations as vehicles for both tractable algorithms and error bounds. The difference is that they concentrate on prediction, not clustering structure. For instance, under the framework of [HM16] one could provide very good guarantees for K-means clustering data that is not clusterable (such as Figure 1 top, left).

## 6   Experimental evaluations for the K-means guarantees

We implemented $\mathrm{SS_{Km}}$ via the SDP solver SDPNAL+[ZST10, YST15]. We also implemented the spectral bound of [Mei06], the only other method offering BTR bounds. The main questions of interest were (1) do our BTRs exist for realistic situations? (2) how tight are the bounds obtained? and (3) given that SDP solvers are computationally demanding, can this approach be applied to reasonably large data sets?

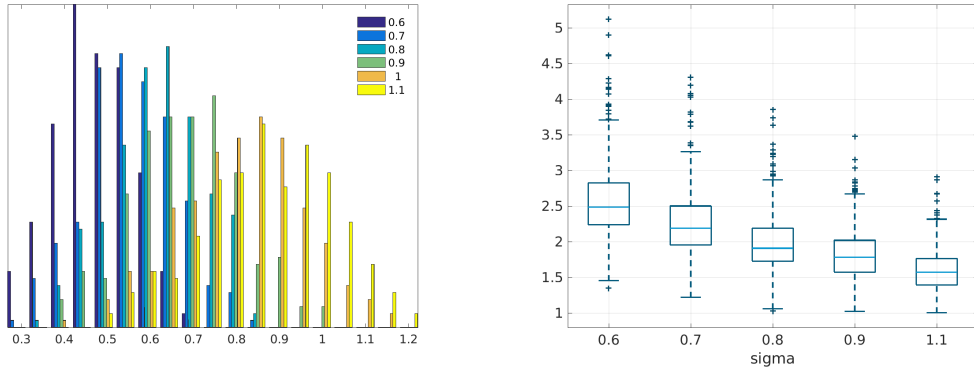

Figure 2: Separation statistics for the $K = 4$ data, $n = 1024$, all $\sigma$ values. Left: histogram of $\min_k \|x_i - \mu_k\|/\min_{k,k'} \|\mu_k - \mu_{k'}\|$ (i.e. distance of point to its center over minimum center separation) colored by $\sigma$. Note that when the clusters are contained in equal non-intersecting balls this ratio is strictly smaller than $0.5$. Right: boxplot of distance to second closest center over distance to own center, versus $\sigma$.

## 6.1 Synthetic data

We sampled data from a mixture of $K = 4$ normal distributions with equal spherical covariances $\sigma^2 I_d$, for $d = 15$ dimensions. The cluster sizes $n_k$ were approximately equal to $\lfloor n/K \rfloor$. The cluster means were at the corners of a regular tetrahedron with center separation $\|\mu_k - \mu_{k'}\| = 4\sqrt{2} \approx 5.67$. The data was clustered by K-means with random initialization, then the bounds $\varepsilon$ and $\varepsilon_{Sp}$ were computed.

In the experiments we also performed *outlier removal*, as follows. For each $x_i$, we computed the sum of the distances to its $p_{\min}/2$ nearest neigbors. We then removed the $n_0$ data points with the largest values for this sum. For good measure, we first added 20 outliers, then removed $n_0 = 4\%n$ respectively $n_0 = 2\%n$ points, depending on whether the $n < 525$ or $n \geq 525$, before computing the bounds $\varepsilon, \varepsilon_{Sp}$. Consequently, these bounds do not refer to all possible clusterings of the original $\mathcal{D}$, but to the "cleaned" dataset. Note however that the outlier removal does not depend on the cluster labels; in fact, we perform it before clustering the data.

Figure 1 displays the bounds $\varepsilon, \varepsilon_{Sp}$ for data with $K = 4, n = 1024$, as well as one instance of the data used. The $\varepsilon$ BTR is much tighter than the spectral bound $\varepsilon_{Sp}$, and, surprisingly enough, holds even when the clusters "touch", i.e when there is no region of low density between the clusters. Otherwise put, the *distribution free* BTR bounds hold even when the data are not contained in non-intersecting balls, which is the best known condition for clusterability *under model assumptions* [ABC$^+$14, ABV14]. Figure 2 (left) shows that, when $\sigma \geq 0.8$, the minimal spheres containing the clusters intersect; on the right we see that there are points which are almost equidistant from two cluster centers.

Next, we performed experiments with unequal cluster sizes $p_{1:4} = 0.1, 0.2, 0.3, 0.4$, and with non-gaussian clusters (details in the Supplement). We also performed experiments with $K = 6$ clusters, with $p_{1:6} = 0.1, 0.18, 0.18, 0.18, 0.18, 0.18$. For $K = 6$ we placed cluster centers along a line (see Figure 3 in the Supplement); this hurts the spectral bound which depends on a stable $K - 1$-subspace, but does not hurt, and may even help the SDP bound $\varepsilon$.

The results are shown in Table 1. The experiments reported in Table 1 are chosen to illustrate the limits of what is achievable by this SS method. In experiments with smaller dispersion values then $0.6$, respectively $0.06$, the bounds $\epsilon$ were very near 0. The table also shows that $\varepsilon$ takes similar values in the case of equal and unequal clusters. However, in the latter case, the condition $\varepsilon \leq p_{\min}/p_{\max}$ is more stringent, hence some of the bounds obtained are not valid.

It is not unexpected that more dispersed clusters, i.e. large $\sigma$, or imbalanced cluster sizes, reflected in a small $p_{min}$, limit the range of the stability guarantees we can obtain. Noise equalizes the Loss landscape, increasing the instability. A bound must be tight enough to "preserve the smallest cluster",

| $K=4$ | Unequal normal clusters | | | Unequal non-normal clusters | | |
|---|---|---|---|---|---|---|
| $\sigma$ | $n=200$ | $n=400$ | $n=800$ | $n=200$ | $n=400$ | $n=800$ |
| 0.6 | 0.00(0.00) | 0.00(0.00) | 0.00(0.00) | 0.001(0.001) | 0.001(0.000) | 0.002(0.007) |
| 0.8 | 0.01(0.01) | 0.01(0.01) | 0.01(0.01) | 0.006(0.004) | 0.004(0.002) | 0.007(0.003) |
| 1.0 | 0.09 (0.05) | 0.06 (0.01) | 0.07 (0.02) | 0.04 (0.02) | 0.03 (0.01) | 0.03 (0.01) |
| 1.2 | 0.28 (0.08) | 0.21 (0.05) | 0.21 (0.03) | 0.16 (0.06) | 0.14 (0.03) | 0.13 (0.03) |

| | $K=6$ clusters | |
|---|---|---|
| | normal | non-normal |
| $\sigma$ | $n=525$ | $n=525$ |
| 0.06 | 0.00(0.00) | 0.005(0.001) |
| 0.08 | 0.01(0.00) | 0.006(0.001) |
| 0.1 | 0.01(0.00) | 0.009(0.003) |

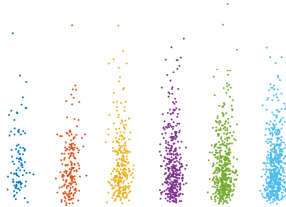

Table 1: BTR bound $\varepsilon$ for $K=4$ (top) respectively $K=6$ (bottom) clusters of unequal sizes (mean and standard deviation over 10 replications). The values in gray are not valid, owing to the fact that $\varepsilon p_{\max} > p_{\min}$ in these cases. Bounds for smaller $\sigma$ values were almost all 0. Bottom, right:the $K=6$, non-normal data for $\sigma=0.1$.

or the clustering is unstable. We see the use of our method in tandem with (existing or future) instability detection methods, such as resampling.

Over all experiments, we have found that the BTR $\varepsilon$ is virtually insensitive to the value of $n$, and degrades slowly when $p_{\min}$ decreases. The main limitation to obtaining BTR is the requirement that $\varepsilon \leq p_{\min}/p_{\max}$; this requirement is based (see the proof of Theorem 2 in the Supplement) on the relationships from [Mei12] between $||X(\mathcal{C}) - X(\mathcal{C}')||$ and $d^{EM}(\mathcal{C}, \mathcal{C}')$. These are not tight, meaning that the regimes in which provable guarantees exist is even larger. From a practical perspective, it is likely that the $\varepsilon$ values marked in gray are valid BTR bounds even though we cannot prove it at this time.

### 6.2 Real data: configurations of the aspirin ($C_9H_8O_4$) molecule

These $n=2118$ samples (see Figure 4 in Supplement) were obtained via Molecular Dynamics (MD) simulation at $T=500$Kelvin by [CTS$^+$17] and represent 3D positions of the 21 atoms of aspirin. It was discovered recently that aspirin's potential energy surface has two energy wells, so we cluster these data with $K=2$, after having removed $n_0 = 0.5\%n = 106$ outliers. The clusters found have relative sizes $p_{\min} = .26, p_{\max} = .74$, and the BTR bound is $\varepsilon = .065$, an informative bound. However, this took over 10h; that encouraged us to try the following heuristic: instead of removing outliers, we removed the 60% of the data points *closest to their centers*. The motivation was that the difficulty of the SDP depends on the cluster boundaries and not on the easy points. The run time reduced to 42 minutes and the bound obtained $\varepsilon = 0.047$ is comparable with the original one. While this speed-up method is ad-hoc, we are confident that it can be made rigurous in the future, opening up the SS method to larger data sets.

## 7  Conclusion

We have introduced a generic method for obtaining distribution free, worst case, guarantees for a variety of clustering algorithms. The method exploits the vast amount of existing work in convex relaxations for clustering; as more results and tighter relaxations appear in this area, the SS method will be able to take advantage of them. For the case of K-means clustering, we have shown empirically that the bounds obtained apply to realistic cases, far surpassing the existing results. It is extremely rare in machine learning to have worst case bounds that are relevant (VC bounds are typically above 1, when they can be computed). However, when the relaxations used by the SS method are tight, we obtain bounds that are not only informative, they are near 0 in non-trivial situations.

The SS method depends only on observed and computable quantities, does not contain undefined constants and does not make any assumptions about the data. However, connections with probabilistic models of the data are possible, and we plan to explore this avenue in future work. The BTR exist only when the data is clusterable. Currently we cannot show that all the clusterable cases can be given guarantees; this depends on the tightness of the relaxation.

The SDP relaxation for K-means has been instrumental to obtaining tight guarantees. In many practical situations, the computational demands of the SDP solver are justified by the guarantees offered. We believe that expanding the method to larger data, beyond the scope of generic SDP solvers, is possible by exploiting the special structure of the SS problem.

Throughout the paper, we have assumed that $K$ is fixed. In practice, $K$ is not known, and it is *chosen after* clusterings with $K = 1, 2, \ldots K_{max}$ have been obtained. Our SS method could replace the (more or less ad-hoc) methods for selecting $K$, with the following: For all $K$ try to find a BTR $\varepsilon(K)$. If successful, the respective clustering and its $K$ are selected. It is of course possible to select more than one $K$, but only if the data indeed supports both clusterings. Thus, indirectly, the BTR can provide a theoretically sound method of selecting $K$.

## Acknowledgments

The author gratefully acknowledges Maryam Fazel for her early interest in this work, for reading a previous version of this paper and for many discussions; the Simons Institute for the Theory of Computing where part of this research was performed, and partial support from the NSF DMS PD 08-1269 and NSF IIS-0313339 awards.

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
