[Supplementary Material · sdp-kmeans-nips18-supplement.pdf]


# How to tell when a clustering is (approximately) correct using convex relaxations

## A Proofs

**Proof of Proposition 1**   $X_{ij} \in [0,1]$ is obvious from the definition (1).

$$\operatorname{trace} X = \sum_{k=1}^{K} \sum_{i \in C_k} X_{ii} = \sum_{k=1}^{K} \sum_{i \in C_k} \frac{1}{n_k} = \sum_{k=1}^{K} n_k \frac{1}{n_k} = K. \tag{12}$$

Denote by $k_0$ the cluster containing data point $i$.

$$(X\mathbf{1})_i = \sum_{j=1}^{n} X_{ij} = \sum_{k=1}^{K} \sum_{i \in C_k} X_{ij} \sum_{j \in C_{k_0}} \frac{1}{n_k} = 1. \tag{13}$$

Moreover, $X = ZZ^T$, hence $X \succeq 0$.                                             $\square$

**Proof of Theorem 2**   Frobenius norm of $X$ is equal to $||X||_F^2 = \sum_{i,j=1}^{n} X_{ij}^2 == \sum_{k=1}^{K} \sum_{i,j \in C_k} X_{ij} = \sum_{k=1}^{K} n_k^2 \left(\frac{1}{n_k}\right)^2 = K$. Then,

$$\begin{aligned} ||Z^T Z'||_F^2 &= \operatorname{trace}(Z^T Z')^T (Z^T Z') = \operatorname{trace}(Z')^T Z Z^T Z' = \operatorname{trace}(ZZ^T)(Z'(Z')^T)(14) \\ &= \operatorname{trace} XX' = \operatorname{trace} X^T X' = <X, X'> \end{aligned} \tag{15}$$

Note also that $||X - X'||_F^2 = ||X||_F^2 + ||X'||_F^2 - 2 <X, X'> = 2K - 2 <X, X'>$. Hence, the optimization problem (SS$_{\text{Km}}$) finds the feasible $X'$ which is furthest away from $X$. This completes Step 1. For Step 2 we can apply the Theorem of [Mei12], which bounds the earthmover distance $d^{EM}$.                                             $\square$

**Proof of Theorem 3**   For any convex problem of the form (4), adding the constraint Loss $\leq l$ and a linear objective preserves convexity. The functions $<X, X'>, <Z, Z'>, <\tilde{X}, \tilde{X}'>$ are obviously linear in the second variable. Hence, the SS1 problem is always convex. Moreover, if (4) has a non-empty relative interior and $X \neq X^*$, the Bound Problem also has a non-empty relative interior, hence strong duality holds. Same arguments hold for $\tilde{X}, Z$. Now, for $X$, is is easy to see from section 3 that the proof Proposition 2 holds regardless of the space $\mathcal{X}$ or of the expression of Loss.

For $Z$, we first notice that $X = ZZ^T$ hence we can prove the result if we can lower bound $||Z^T Z'||_F^2$ for any pair of clusterings. We have $<Z, Z'> = \operatorname{trace} Z^T Z'$. Now, for any symmetric matrix $A$ with non-negative elements, $||A||_F^2 = \operatorname{trace} A^2 = \sum_{i \in [n]} \lambda_i(A)^2 \geq \frac{1}{2}(\sum_{i \in [n]} \lambda_i(A))^2 = \frac{1}{2}(\operatorname{trace} A)^2$. Let $\delta' = \delta^2/2$. Then $\varepsilon = (K - \delta')p_{\max}$ is a BTR whenever it is smaller or equal to $p_{\min}$, by an argument similar to the proof of Proposition 2.

For the $\tilde{X}$ representation, we note that $||\tilde{X}||_F^2 = \sum_{k \in [K]} n_k^2 \leq (n - K + 1)^2 + (K - 1)$ for any $\tilde{X}$ representing a clustering. Hence, $||\tilde{X} - \tilde{X}'||_F^2 = ||\tilde{X}||_F^2 + ||\tilde{X}'||_F^2 - 2 <\tilde{X}, \tilde{X}'> \leq ||\tilde{X}||_F^2 + (n - K + 1)^2 + (K - 1) - 2\delta$. We now apply Theorem 27 of [Mei12] which states that $d^{EM}(\mathcal{C}, \mathcal{C}') \leq \frac{1}{2n^2 p_{\min}} ||\tilde{X}(\mathcal{C}) - \tilde{X}(\mathcal{C}')||_F^2$ and obtain $\varepsilon = \frac{\sum_{k \in [K]} n_k^2 + (n-K+1)^2 + (K-1) - 2\delta}{2p_{\min}}$ whenever $\varepsilon \leq p_{\min}$.                                             $\square$

## B BTR from the K-means LP relaxation

We present these to further illustrate the versatility and ease of use of our method. In [ABC$^+$14] the following relaxation to the K-means problem is presented.

$$\delta_{\text{LP}}(\mathcal{C}) = \min_{X \in \mathcal{X}_{\text{LP}}} \langle D, X \rangle. \tag{16}$$

Figure 3: Some data used in the experiments. In the first three plots, the clusters are sampled from mixtures of spherical Gaussians. In the last, one of the 15 coordinates is from a $\mathrm{Gamma}(2, .4)$ distribution and rescaled by $\sigma$. Separation is the distance between the Gaussian means, and $\sigma$ is the standar deviation of the Gaussians. The $K = 6$ data sets are designed to be hard for the spectral bounds but not for the SDP bounds.

In the above, the mapping $\mathcal{C} \to X(\mathcal{C})$ is the same as in (1), but the convex set $\mathcal{X}_{\mathrm{LP}}$ over which the relaxation is done is $\mathcal{X}_{\mathrm{LP}} = \{X \succeq 0, \, \mathrm{trace}\, X = K, \, X\mathbf{1} = \mathbf{1}, \, X_{ij} \leq X_{ii} \text{ for all } i, j \in [n], \, X_{ij} \in [0, 1] \text{ for all } i, j \in [n]\}$. This relaxation can be cast as an LP, making it more attractive from the computational point of view. It is straightforward to state Sublevel Set problem SS1.

$$\delta_{\mathrm{LP}}(\mathcal{C}) \;=\; \min_{X' \in \mathcal{X}_{\mathrm{LP}}} \langle X(\mathcal{C}), X' \rangle, \quad \text{s.t. } \langle D, X' \rangle \;\leq\; \langle D, X \rangle. \tag{17}$$

Since the original problem can be cast as an LP, (17) can too. Moreover, since $\delta_{\mathrm{LP}}(\mathcal{C})$ bounds the same quantity $< X(\mathcal{C}), X' >$, Proposition 2 applies as well. If bounds from multiple relaxations are obtained, the tightest one bounds the distance $d(\mathcal{C}, \mathcal{C}')$.

In [ABC$^+$14] it is shown that, for data generated from separated balls, the SDP relaxation is strictly tighter than the LP relaxation. This suggests that the BTR from the LP (17) will not be as tight as the SDP bounds from 2.

## C   Additional experimental results

Figure 4: The first two principal components of the 57-dimensional aspirin data. The data is a Molecular Simulation sequence of 211762 configurations. We sample every 100-th point of the data for clustering. The axes are represented *at scale*.

Figure 5: Results for the $K = 4$ data, $n = 256$, over 10 replications.

| | Unequal normal clusters | | | Unequal non-normal clusters | | |
|---|---|---|---|---|---|---|
| $\sigma$ | $n = 200$ | $n = 400$ | $n = 800$ | $n = 200$ | $n = 400$ | $n = 800$ |
| 0.06 | 11.72 (7.03) | 129.72 (68.32) | 1172 (954) | 18.93 (13.31) | 78.59 (52.21) | 1185.89(678.90) |
| 0.08 | 95.38(47.06) | 433.81(134.52) | 3396 (890) | 108(72) | 307(111) | 3600(1185) |
| 0.10 | 54.92(26.10) | 487.98(129.72) | 3298(1016) | 103(78) | 542(169) | 4185(693) |
| 0.12 | 21.95 (9.53) | 154.42 (52.40) | 1410 (282) | 69.04(46.59) | 227(69.68) | 1936(545) |

Table 2: Run time for the experiments wih $K = 4$ clusters of unequal sizes (mean and standard deviation over 10 replications) from Table 1.