[Reviews · NeurIPS 2018]

Reviewer 1



This paper presents a general method to derive bounds on how close a given clustering is to an optimal clustering where optimal is defined according to certain loss functions. The method can be used for any clustering loss function for which convex relaxation exists (e.g., K-means, spectral clustering). They show in experiments they obtain much better bounds than the only related work (as far as I know) on K-means [Mei06]. The paper is well-written and easy to follow and addresses an important problem of evaluating the quality of clusterings. The main contribution of the paper is to make use of tighter convex relaxations to derive bounds on the distance between optimal clustering and a given clustering. It is obvious that tighter the convex relaxation the better the bounds are and hence they use SDP relaxation for the K-means problem (which can be computationally prohibitive for large datasets). Usually the convex relaxation already gives a bound on how far the optimal loss value is from the loss of a given clustering. Is this already indicative of how far the optimal solution is from the given clustering? My main concern is how good the bounds are in more realistic settings. Moreover, the method can fail to get a valid bound if some certain technical condition fails (which is easy to check after solving the convex relaxation). As they show in experiments the method fails to obtain valid bounds for difficult settings (high noise, unequal sizes of clusters). On the positive side, + the method provides much better bounds than the spectral bounds derived in [Mei06] + several synthetic experiments are conducted under various settings to illustrate the bounds obtained by the method. Comments: - From Theorem 3 it appears that the method works for spectral clustering as well; experiments on spectral clustering would be interesting, e.g., to check the quality of the solutions obtained by recent methods like [HeiSet11, RMH14]. - It was not clear the need for removing outliers in the synthetic experiments. - The method is computationally expensive; they hint at reducing the complexity by removing data points that do not affect the clustering but this needs to be further formalized. - More experiments on real world data would enhance the paper (currently only one real world data is used). - Missing references: line 248, line 253 [HeiSet11] Beyond spectral clustering: Tight continuous relaxations of balanced graph cuts. NIPS 2011.

Reviewer 2



This paper combines convex optimization methods with clustering under stability assumptions. It directly incorporates stability-motivated conditions into the SDP formulation for k-means, and proves several guarantees based on such relaxations. It also experimentally measures the distances of nearly-optimum solutions to a point, and shows that practical data sets do in fact have such stability properties. I am not familiar with the vast literature on clustering, or even recent works involving stability assumptions. However, I believe the design of objectives with such assumptions in mind is natural, and that the paper did significant work in such direction. This includes both the bounds proven, and the experimental studies involving real data sets as well as additional heuristics. I believe these are significant contributions that may motivate further work on convex relaxations under stability assumptions, and would like to advocate acceptance of this paper.

Reviewer 3



This paper introduces a general template, based on convex relaxation, for post-clustering validation that has theoretical guarantee, without assuming clusterability on the dataset. The authors provides an example of how in the context of k-means clustering, this can be done via SDP formulation. Although there are some parts where I feel the paper can be improved, in general I think this is an interesting theoretical contribution to an area that is important for practical concern (validation for clustering). Here are some detailed comments for the authors to consider: - Some definitions are confusing: For example, the way the authors define an optimization problem is a bit confusing; in (5) and (6), it took me a while to understand that epsilons are not pre-defined constraints but the optimal values achieved. Also, what exactly is the matrix norm used here to gauge the difference X-X'? - Is (6) ever tractable? Isn't X* the optimal value unknown to the algorithm? Also, it doesn't seem to appear in the subsequent text (why not removing it?) - How is (12) a special case of (5)? (again, what exactly is the matrix norm used in (5))? - Similarly, in Theorem 3, inner product is used to measure the distance (similarity) between matrices. How is it related to (5)? - Section 4 is rather unclear. Can the authors spend a few words discussing what exactly are the clustering paradigms (losses) used in the referred papers? - If the method can be indeed extended easily to clustering objectives other than k-means (as claimed in Sec 4), then perhaps the authors should strengthen this statement by showing empirical results.